# Quantitative Proteomic Study Unmasks Fibrinogen Pathway in Polycystic Liver Disease

**DOI:** 10.3390/biomedicines10020290

**Published:** 2022-01-27

**Authors:** Adrian Cordido, Marta Vizoso-Gonzalez, Laura Nuñez-Gonzalez, Alberto Molares-Vila, Maria del Pilar Chantada-Vazquez, Susana B. Bravo, Miguel A. Garcia-Gonzalez

**Affiliations:** 1Group of Genetics and Developmental Biology of Renal Diseases, Nephrology Laboratory (N°11), Health Research Institute of Santiago de Compostela (IDIS), Santiago de Compostela Clinical Hospital Complex (CHUS), 15706 Santiago de Compostela, Spain; adriancosman@hotmail.es (A.C.); martavizosoglez@gmail.com (M.V.-G.); lauranugo82@gmail.com (L.N.-G.); 2Genomic Medicine Group, Santiago de Compostela Clinical Hospital Complex (CHUS), 15706 Santiago de Compostela, Spain; 3Biostatistics Platform, Health Research Institute of Santiago de Compostela (IDIS), Santiago de Compostela Clinical Hospital Complex (CHUS), 15706 Santiago de Compostela, Spain; amolares@hotmail.com; 4Proteomic Platform, Health Research Institute of Santiago de Compostela (IDIS), Santiago de Compostela Clinical Hospital Complex (CHUS), 15706 Santiago de Compostela, Spain; mariadelpilarchantadavazquez@gmail.com; 5Galician Public Foundation of Genomic Medicine, Santiago de Compostela Clinical Hospital Complex (CHUS), 15706 Santiago de Compostela, Spain

**Keywords:** PLD, SWATH, quantitative proteomics, therapeutic targets

## Abstract

(1) Background: Polycystic liver disease (PLD) is a heterogeneous group of congenital disorders characterized by bile duct dilatation and cyst development derived from cholangiocytes. Nevertheless, the cystogenesis mechanism is currently unknown and the PLD treatment is limited to liver transplantation. Novel and efficient therapeutic approaches are th6us needed. In this context, the present work has a principal aim to find novel molecular pathways, as well as new therapeutic targets, involved in the hepatic cystogenesis process. (2) Methods: Quantitative proteomics based on SWATH–MS technology were performed comparing hepatic proteomes of *Wild Type* and mutant/polycystic livers in a polycystic kidney disease (PKD) murine model (*Pkd1^cond/cond^*;*Tam-Cre^−/+^*). (3) Results: We identified several proteins altered in abundance, with two-fold cut-off up-regulation or down-regulation and an adjusted *p*-value significantly related to hepatic cystogenesis. Then, we performed enrichment and a protein–protein analysis identifying a cluster focused on hepatic fibrinogens. Finally, we validated a selection of targets by RT-qPCR, *Western blotting* and immunohistochemistry, finding a high correlation with quantitative proteomics data and validating the fibrinogen complex. (4) Conclusions: This work identified a novel molecular pathway in cystic liver disease, highlighting the fibrinogen complex as a possible new therapeutic target for PLD.

## 1. Introduction

Polycystic liver disease (PLD) is a heterogeneous group of congenital disorders characterized by bile duct dilatation and cyst development derived from cholangiocytes (epithelial cells from bile duct). The numerous cysts spread and progress throughout the liver parenchyma compromising its function [1]. PLDs are inherited in dominant or recessive form and can occur in isolation (isolated polycystic liver disease or PLD) or, more commonly, as an extra-renal manifestation of autosomal dominant adult polycystic kidney disease (PKD) [2]. In autosomal dominant polycystic kidney disease (ADPKD), ~85% of the patients develop liver cysts throughout hepatic parenchyma and its severity can range from a few cysts to a severe hepatic cystogenesis [3,4]. Meanwhile, in autosomal recessive polycystic kidney disease (ARPKD), liver cysts could develop, but the principal hepatic complication is a defective remodeling of the ductal plate with congenital hepatic fibrosis [5,6]. For both, PLD is the principal extrarenal manifestation and requires clinical management and treatment [2,5]. The principal clinical approach for PLD patients is to reduce their clinical symptoms. Management of bile duct complications is the main clinical feature to be controlled, and the final treatment strategy is usually liver transplantation [7], since there is no effective pharmacological treatment for PLD [2,5]. Multiple molecular pathways have been associated with the disease [8], such as increased fluid secretion [9,10], proliferation [11], fibrosis [12,13], ciliary defects [14] and others [7,8]; but the main mechanism undergoing cystogenesis remains to be elucidated. The identification of new potential therapeutic targets is urgently required. All these studies were based on molecular studies using different classical molecular tools: 3D cell culture [7,8], immunohistochemistry [7,8,9,11,12], ELISA [9,11], transmission electron microscopy (TEM) [12], immunocytochemistry [11], RT-qPCR (quantitative real-time PCR) [7,8,10,11] and Western blot [7,8,10,11,12]. Approaching the study of disease from new perspectives, well-established preclinical animal models, and taking advantage of new technologies, can be a benefit in the discovery of new molecular mechanisms of disease to translate molecular discoveries into new therapies.

Liquid chromatography coupled to tandem mass spectrometry (LC–MS/MS) has been the most commonly used technology for the characterization of disease proteomes as well as the identification of possible disease biomarkers or drug screening [15,16]. An emerging strategy named SWATH–MS (sequential window acquisition of all theoretical fragment-ion spectra–mass Spectrometry) allows a reproducible, reliable, high-throughput quantification with up to thousands of proteins analyzed per biological sample, and with the possibility of simultaneously performing a large number of assays [16,17]. For this technology, the samples are digested with trypsin and analyzed by liquid chromatography coupled to a tandem mass chromatography works in the so-called Data-Independent Acquisition (DIA) mode [16] (see Material and Methods). Next-generation proteomics SWATH–MS technology allows quantifying the proteome and identifying differences in protein abundance in different samples and conditions.

In the present study, we performed a differential proteomic quantitative analysis based on SWATH–MS technology in polycystic liver disease from mutant animal models. We report a list of proteins with statistical significance of differential protein abundance, which were characterized by different enrichment analysis and data curation. Finally, we validated the SWATH data by identifying and highlighting the fibrinogen complex as a new mechanism of disease and possible new therapeutic target for PLD.

## 2. Materials and Methods

### 2.1. Murine Model

In this study, we used a murine *Pkd1* conditional-knockout animal model, the C57/BL6 k *Pkd1^cond/cond^*;*Tam-Cre* [18,19]. Cre- mouse model was used as Wild Types (WT), and the Cre^+^ as Mutant (KO). Genotypes were confirmed by standard PCR, following the conditions and primers described [18]. Deletion of *Pkd1* was always at postnatal days 10 and 11 (p10 and p11) with a single intraperitoneal injection of tamoxifen (10 mg/40 g) (Sigma^®^, St. Louis, MO, USA No. T5648), diluted in corn oil (Sigma–Aldrich^®^, St. Louis, MO, USA No. C8267), both days in the mother of the litters to be induced. The animals were euthanized at postnatal day 30 (p30). The groups were established by different individuals and organized randomly (randomization) and keeping 1:1 ratio between males and females in all groups. Anatomical characteristic features were used to identify the sex of the animals. The mice were housed in a pathogen-free facility (SPF) in accordance with the established conditions of the University of Santiago.

### 2.2. Protein Extraction and Digestion

At the p30 sacrifice point, the dissected livers were stored at −80 °C. The whole liver was ground in liquid nitrogen using a liquid nitrogen-cooled mortar (DD Biolab^®^, Barcelona, Spain No. 088763), with half of the tissue assigned to protein extraction and the other half to RNA extraction. Protein extracts were prepared with RIPA lyses buffer (10 Mm TRIS, 5 mM EDTA, 150 mM NaCl, 0.1% SDS, 1% TRITON X-100 and 1% sodium deoxycholate). 1% protease and phosphatase inhibitors (Sigma^®^ No. P8340 y No. P0044) were also added. This tissue lysate underwent a centrifugation process (for 30 min at 4 °C and 14,000 rpm). The supernatant was then recollected and quantified by the Bradford protein assay (Bio-Rad^®^ No. 5000001). Protein aliquots (100 µg) were concentrated in a single band in a 10% sodium dodecyl sulphate-polyacrylamide gel electrophoresis (SDS-PAGE), cut and submitted for manual digestion as previously described [20]. Finally, the extracted peptides were dissolved in 0.1% formic acid for further analysis.

### 2.3. Proteomics

#### 2.3.1. DDA-Data Dependent Analysis

4 µL of each sample (over 4 µg of protein) were analyzed by shotgun data-dependent acquisition (DDA) approach by micro-LC–MS/MS. The samples were separated by the Ekspert nLC425 micro- LC system (Eksigen^®^, Dublin, CA, USA) using a YMC-TRIART C18 trap column with a 3 mm particle size and 120 Å pore size (YMC Technologies, Teknokroma, Barcelona, Spain) and a column Chrom XP C18 150 mm × 0.30 mm, 3 mm particle size and 120 Å pore size (Eksigen^®^) at a flow rate of 10 μL/min. The solvents used were solvent A (water, 0.1% formic acid) and solvent B (ACN, 0.1% formic acid). The gradient was from 5% to 95% B for 30 min, 5 min at 90% B and, finally, other 5 min at 5% B for column equilibration, for a total time of 40 min. The mass spectrometer coupled was a hybrid quadrupole-TOF mass spectrometer, 6600 (SCIEX^®^, Framingham, MA, USA) operating with a data dependent acquisition system in positive ion mode. Using the mass spectrometer, a 250 ms survey scan was performed at 400 to 1250 *m*/*z* followed by MS/MS experiments at 100 to 1500 *m*/*z* (25 ms of acquisition time) for a total cycle time of 2.8 s. Fragmented precursors were added to the dynamic exclusion list for 15 s, any ion with charge +1 was excluded from the MS/MS analysis. MS raw file and database searches were combined and performed using ProteinPilot software v.5.0.1. (SCIEX^®^, Framingham, MA, USA) using a mouse-specific UniProt Swiss-Prot database. It is necessary to specify iodoacetamide cysteine alkylation as fixed modification and trypsin digestion. The false discovery rate (FDR) was set to 1 for peptides and proteins with a confidence score greater than 99% [21].

#### 2.3.2. Generation of the References Spectral Library

The spectral library was performed by a DDA method as described above, but instead each independent sample, a pool of each group was performed with 4 µL of each sample (over 4 µg of protein) were run in order to obtain more representative protein identification in the spectral library. All raw files were launched together into the Uniprot database to obtain the spectral library using the protein pilot conditions mentioned in the previous section.

#### 2.3.3. Quantification by SWATH and Data Analysis

The SWATH–MS acquisition was performed using a DIA method. We used 4 groups with 4 biological replicates per group and 3 technical replicates per sample (see Appendix A for more details). The tissues were analyzed and, in all cases, compared in *Wild Type* (from *Pkd1^cond/cond^*;*Tam-Cre^−^* mouse) and Mutant (from *Pkd1^cond/cond^*;*Tam-Cre^+^* mouse) conditions. 4 µg of protein per sample and technical replicates were run using a SWATH method. For each set of samples, the width of the 100 variable windows was optimized according to the ionic density found in the library DDA runs using a Sciex SWATH variable window spreadsheet. Therefore, the SWATH method was based on a cycle of repetitions that consisted of the acquisition of 100TOF MS/MS scans (400 to 1500 *m*/*z*, high sensitivity mode, 50 ms acquisition time) of isolation windows of sequential overlapping precursor variables with widths (1 *m*/*z* overlap) covering the 400 to 1250 *m*/*z* mass range with a previous TOF MS scan (400 to 1500 *m*/*z*, 50 ms acquisition time) for each cycle. Total cycle time was 6.3 s.

#### 2.3.4. Data Analysis

All proteins in the ion library that were identified by ProteinPilot with an FDR below 1% were quantified by the SWATH method. Once individual samples were acquired, the spectral alignment and targeted data extraction were performed by PeakView v.2.2. (SCIEX^®^, Framingham, MA, USA) matching the reference spectral library as described below [22,23,24,25,26]. The retention times of the peptides that were selected for each protein, were realigned in each run according to the iRT peptides present in each sample and were eluted along the whole-time axis. The extracted ion chromatograms were then generated for each ion of the selected fragment; the peak areas for the peptides were obtained by adding the peak areas of the ions of the corresponding fragments. PeakView computed an FDR and a score for each assigned peptide according to the chromatographic and spectra components; only peptides with an FDR below 5% were used for protein quantitation. Similarly, to obtain the areas for protein quantification as a function of signal intensity, up to 10 peptides per protein and seven fragments per peptide were selected. All shared and modified peptides were excluded from processing. Five-minute windows and 30 ppm widths were used to extract the ion chromatograms.

Integrated area peaks (processed “.mrkvw” files from PeakView, 22 December 2017) were exported directly to MarkerView software (SCIEX^®^, Framingham, MA, USA) for relative quantitative analysis. The export generates three files that contain quantitative information about individual ions, the summed intensity of different ions for a particular peptide and the summed intensity of different peptides for a particular protein [27,28,29,30]. MakerView uses processing algorithms that accurately find chromatographic and spectral peaks directly from raw SWATH data. MarkerView data alignment compensates for minor variations in both mass and retention time values, ensuring that identical compounds in different samples are accurately compared to each other. To control the possible uneven loss of samples in the different samples during the sample preparation process, we performed a TAS (total area sum) normalization [31,32,33]. Version R 3.6.2 (https://www.r-project.org/, 12 February 2020) and a set of packages has been used for the analysis [34]. Volcano plots were used to graphically visualize the results that were generated by plotting the log(2)-fold changes for all proteins identified against their −log(10) *p*-value. Unsupervised cluster analysis heatmap figures of proteins with significant differences according to SWATH-MS analysis were generated to detect different cluster of proteins in different studied groups.

#### 2.3.5. Signaling Pathway Analysis

Different functional analysis was performed using differentially expressed proteins in order to evaluate the most relevant interaction networks, related pathways or cellular components and other associated proteins or pathways. UniProt protein accession numbers (https://www.uniprot.org/, 28 August 2021) were used. Proteins with differences in protein abundance were subjected to: String^®^ (https://string-db.org/, 25 October 2021) using Gene Ontology (GO) terms and FunRich^®^ (http://funrich.org/index.html, 21 October 2021) to study protein networks and biological association between proteins; and to Reactome^®^ (https://reactome.org/, 25 October 2021) to study the pathways involved in relation or interaction with significantly expressed proteins.

### 2.4. Histology and Measurement of Cystic Index

The largest lobe of the liver of the animals was always fixed in paraformaldehyde (4% paraformaldehyde buffered solution, pH = 7, Labbox No. FORM-D0P-10K) overnight at 4 °C. The livers were then dehydrated with xylene and ethanol and finally embedded in paraffin. Cross sections of 4.5 µm were collected. For hematoxylin–eosin (HE) and Masson trichrome staining’s standard protocols was used.

For quantifications, slides of longitudinal sections of the largest lobe of liver were always taken as representative images and, at least six different sections per animal, visualized under an Olympus BX51 microscope connected to an Olympus Camera DP70, were used for quantification of cystic index and number of cysts. The quantification of both parameters was calculated with Automatic Cyst Recognition software tool called CystAnalyser (https://citius.usc.es/transferencia/software/cystanalyser, 1 July 2020) as previously described [35].

### 2.5. Clinical Chemistry

Blood was collected before euthanasia, and serum was obtained using BD Vacutainer SST II Advance tubes. Serum alanine transaminase (ALT) and alkaline phosphatase (ALP) were measured using the ADVIA 2400 Chemistry System (Siemens Healthineers, Erlangen, Germany).

### 2.6. RNA Extraction

RNA was isolated as previously described [36] by using TRIzol protocol (TRI Reagent, Sigma^®^, St. Louis, MO, USA, No. T9424), in which chloroform (Sigma^®^ No. C2432) served as denaturing agent for the cells and isopropanol (Sigma^®^ No. 650447) was the precipitating agent for RNA. The precipitate was diluted in DEPC water (Ambion^®^,Austin, TX, USA, No. AM9920), which also contains 1% SUPERaseIn RNase Inhibitor (Thermo Fisher^®^, Waltham, MA, USA, No. AM2694).

### 2.7. Real-Time Quantitative PCR

One microgram of total liver RNA was treated with DNase I (Invitrogen^®^, Carlsbad, CA, USA, No. 18068015). Then, reverse transcription was performed with SuperScript^®^ III First-Strand Synthesis System kit (Invitrogen^®^, No. 18080051), and gene expression analysis by quantitative real-time PCR using FastStart Universal SYBR GREEN Master (ROX) (Roche^®^ No. 04913914001) in a Mx3005P system (Agilent Technologies^®^, Santa Clara, CA, USA). The primers used for this study were designed with Primer 3 Plus (http://tiny.cc/pf928y, 11 March 2020) and are based on the cDNA reference sequences published in Ensembl genome browser (http://www.ensembl.org/index.html, 11 March 2020). The full sequences of all primers used in the study are found in Appendix A. The following program was used for all complete PCR reactions: a denaturation step with 30 s at 95 °C; 45 cycles of 30 s at 95 °C, 1 min at 60 °C and 30 s at 72 °C; followed by a cooling step. As we have previously carried out [36], each sample (*n* = 6 for each group) was run in triplicate in each experiment. The results were analyzed through the absolute quantification method. Consequently, a standard curve with known concentrations was used in the assay and, in addition, each sample was normalized to a constantly expression gene. In this study, the *Gapdh* gene (glyceraldehyde-3-phosphate dehydrogenase) was used as the housekeeping gene.

### 2.8. Western Blotting

Western blotting assays were performed on whole liver lysates. Total protein content was determined using Bradford protein assay (Bio-Rad^®^, Hercules, CA, USA, No. 5000001) and the same amount of each protein (20–30 µg) was boiled at 95 °C for 5 min and separated in 10% SDS-PAGE under reducing conditions. The PageRuler^TM^ Prestained Protein Ladder (Thermo Fisher^®^, Waltham, MA, USA, No. 26616) was used as the protein molecular weight standard. Subsequently, the proteins were transferred to nitrocellulose membranes (Bio-Rad^®^ No. 1620115). To avoid nonspecific binding, the membranes were blocked for 1 h at room temperature using a 2.5% BSA solution (NZYTech^®^, Lisboa, Portugal, No. MB046) for phosphorylated proteins, or SuperBlock Blocking Buffer (Thermo Fisher^®^, Waltham, MA, USA, No. 37515). Subsequently, the membranes were incubated overnight at 4°C with primary antibodies and next washed in 0.3% Tween-20 in PBS. Finally, the nitrocellulose membranes were incubated for 1 h at room temperature with the HRP-linked secondary antibodies. Signals were developed with the SuperSignal™ West Pico PLUS Chemiluminescent Substrate kit (Thermo Fisher^®^, Waltham, MA, USA, No. 34580) or PierceTM ECL Western Blotting Substrate kit (Thermo Fisher^®^, Waltham, MA, USA No. 32109) prior to visualization in the ChemiDoc™ Imaging System (Bio-Rad^®^, Hercules, CA, USA). Protein bands were quantified using ImageJ^®^ Lab software (v 4.0.1, https://imagej.nih.gov/ij/, 26 March 2021). Protein densitometry levels were normalized to housekeeping proteins GAPDH or β-ACTIN. For each group. *N* = 6 was used, in all experiments.

The following primary antibodies w”re u’ed for *Western blotting*: anti-ANXA2 (1:500, Santa Cruz Biotechnology Inc.^®^, Dallas, TX, USA #sc-28385), anti-SAMP (1:500, Santa Cruz Biotechnology Inc.^®^, Dallas, TX, USA, No. sc-393948), anti-FIBB (1:500, Santa Cruz Biotechnology Inc.^®^, Dallas, TX, USA, No. sc-271035), anti-FIBA (1:500, Santa Cruz Biotechnology Inc.^®^, Dallas, TX, USA, No. sc-398806), anti-FIBG (1:500, Santa Cruz Biotechnology Inc.^®^, Dallas, TX, USA, No. sc-133157), anti-HAO2 (1:1000, Thermo Fisher^®^, Waltham, MA, USA, No. 113442), anti-GAPDH (1:1000,Cusabio^®^, Houston, TX, USA, No. CSB-RA009232A0HU) and anti-β-ACTIN (1:1000, Cell Signaling Technology^®^, Danvers, MA, USA, No. 8457), as well as the secondary antibody Goat anti-Rabbit IgG HRP (1:5000, Thermo Fisher^®^, Waltham, MA, USA, No. 31460) or Goat anti-Mouse IgG HRP (1:5000, Thermo Fisher^®^, Waltham, MA, USA, No. 31430).

### 2.9. Immunohistochemistry and Quantification

For immunohistochemistry, sodium citrate buffer pH = 6 (Agilent Dako^®^, Glostrup, Denmark No. S2369) was used for antigen retrieval and tissue was blocked with antibody diluent (Agilent Dako^®^ No. K8006). The antibody used for immunohistochemical analysis was fibrinogen (Agilent Dako^®^ No. A0080), and HPR-conjugated detection system for the immunodetection. Finally, immunohistochemistry’s quantification was performed using the “Split channels” function (green channel) from Fiji-ImageJ^®^ software (https://imagej.net/software/fiji/, 13 October 2021). For each sample, *n* ≥ 6 images was used.

### 2.10. Statistics Analysis

Data are presented as means ± SEM for all cases (bar or point plots). First, normal distribution was confirmed using the Kolmogorov–Smirnov test. Two-tailed *t*-test was used to determine significance of differences between two groups. *p* < 0.05 was considered statistically significant. Furthermore, ** corresponds to *p*-values < 0.01 and *** to *p*-values < 0.001. All datasets were analyzed using GraphPad Prism software (version 9, https://www.graphpad.com/, 13 October 2021).

For proteomic studies, a Student’s *t*-test (using MarkerView software, 22 December 2017) was performed to compare the groups in pairs in order to identify proteins, which were significantly differentially represented using adjusted *p*-value < 0.05 (multiple correction FDR method) and fold change >2 as cut-off. Unsupervised multivariate statistical analysis using principal component analysis (PCA) was performed to compare the data across the samples.

## 3. Results

### 3.1. Kidney and Liver Does Not Follow the Same Cystic Mechanisms

Previous studies showed that early inactivation of the *Pkd1* gene (before p12) triggers rapid development of polycystic disease (cystic window), whereas inactivation after p14 leads to late cyst formation after 4–5 months, with a mild phenotype (non-cystic window) [19]. Using the same orthologous model of ADPKD (*Pkd1^cond/cond^;Tam-Cre*) [18,19], we investigated whether this window of renal development corresponded in time with a similar window of hepatic development. To draw a direct comparison with our previous studies [37], mice were induced with tamoxifen to inactivate the *Pkd1* gene at postnatal days 14 (p14) and 12 (p12) and sacrificed at the age of 30 days (p30). Interestingly, we observed a cystic phenotype at both time points with a different degree of disease (mild and severe), suggesting that the kidney and liver have independent cystic developmental windows (Figure 1a), and/or possible different timings and progression of disease. No differences were found between males and females at this slaughter age between both windows of inactivation (*n* = 6 was used for each group of mutants). The p14 mutant animals presented a milder phenotype than p12 mutant group with lower values of cystic index, number of cysts and hepatic function according to ALP serum value (Figure 1b–d). This is the first study showing different developmental windows/mechanisms for *Pkd1* liver and kidney deficiency. Those molecular differences will have to be determined in future studies.

### 3.2. Shotgun and SWATH–MS Proteomic Analysis in PLD

As referred in Figure 2, we performed differential proteome analysis at both cystic stages (p12—severe cystic disease- and p14—mild cystic disease-) of Mutants (KO) in comparison to *Wild Type* (WT) animals, in order to identify and characterize relevant molecular mechanisms undergoing liver cystogenesis and disease progression.

#### Protein Expression Pattern in Hepatic Cystogenesis

First, we performed proteomic mass spectrometry analysis by shotgun data-dependent acquisition DDA-MS or DDA. This analysis allows characterizing the proteome from complete and individual samples, providing the presence/absence of proteins with high sensitivity but without providing their quantification. We exclusively considered all the proteins expressed differentially in more than five independent samples from a total of six samples, for p14 and p12 stages of *Wild Type* and mutant individuals, respectively (Appendix A). At p14, 1 WT-exclusive (or down-regulated for the disease) and 7 MUT-exclusive (up-regulated) proteins were identified, while at p12, eight WT-exclusive and six MUT-exclusive proteins were observed (Appendix A).

Next, we used a method that allowed us to quantify differentially expressed proteins. We performed a SWATH–MS quantitative proteomic analysis on the very same samples that we used for DDA, obtaining ~1500 proteins per sample and more than 2000 proteins in each study group. First, we performed total area sum (TAS) normalization to evaluate that our samples followed a normal distribution pattern, as shown in Appendix A. Next, we studied the protein with significant differences between *Wild Type* and Mutant groups. Table 1 shows the proteins with a significant change in abundance with a two-fold increase (up-regulated) or decrease (down-regulated), and a significant adjusted *p*-value (according to the parametric Student’s *t*-test) in WT vs MUT samples at p14 and p12. In total, six proteins showed significant differences in protein abundance between the WT-p14 and MUT-p14 groups (five up-regulated proteins and one down-regulated proteins). Between WT-p12 and MUT-p12, 26 proteins (20 proteins up-regulated and 6 proteins down-regulated) showed significant differences (Figure 3a and Table 1). Graphically, these variations can be observed through volcano plots, which were generated by plotting the log(2)-fold changes for all proteins identified against their −log(10) *p*-value (Figure 3b). Significant differences among protein abundance levels can be visualized in the heat map with individualized values according to the areas of the spectral library, and its clustering level in the cluster heat map analysis (Figure 3c and Appendix A). These clusters detected by the SWATH–MS analysis help us to identify those qualitatively separated samples, in order to be considered for the quantitative analysis (Appendix A). An unsupervised multivariate statistical analysis was performed using principal component analysis (PCA) to compare the data between samples (Appendix A). The heatmap and PCA data demonstrated reproducibility among the sample triplicates as the three replicates closely clustered in both analyses. We can observe both in the PCA and in the heatmap cluster analysis (Appendix A) how some samples did not group correctly, overlapping between both clusters. Nonetheless, we can see the tendency to separate data in *Wild Type* and Mutant clusters. The lower number of proteins with significant differences in p14 vs p12 could be justified by the different degree of severity (Figure 2), suggesting that the number of proteins and pathways increases based on the severity and disease status of the cystic phenotype.

### 3.3. Clustering, Pathway Enrichment and Protein–Protein Interaction Analysis of Gene Expression in PLD

To establish the function related to the differentially expressed proteins identified by SWATH-MS analysis, we used several bioinformatic tools and forms of analyses. Gene Ontology (GO) terms and pathways analyses were performed using FunRich [39,40], String [41] and Reactome [42]. The proteins were sorted in FunRich gene enrichment analysis. Regarding the biological process, the most enriched group of proteins for p14 group (mild cystic) were related to fatty acid transport and prostaglandin biosynthesis, and for p12 (severe cystic) group with fibrinogen/fibrin activity, cell–cell/matrix adhesion and metabolic process (Figure 4a, upper panel). ANXA2 and the fibrinogen complex were the two most enriched cellular components in p14 and p12 groups, respectively. Furthermore, the extracellular space was the cellular component with the highest percentage of proteins in both groups (Figure 4a, lower panel). Similarly, the same analysis was established with string analysis, identifying different metabolic processes as the main biological process, and the extracellular space as the most enriched cell component; consistent with GO results (Appendix A). Finally, we repeated the analysis with the Reactome platform, finding that metabolism and immune system were the most enriched pathways (Appendix A).

Protein–protein or cluster analysis based on string analysis was also performed in order to investigate possible protein–protein interactions (interactions with experimental evidence). Numerous protein interactions were identified in different groups (Appendix A), but one cluster at p14 and another at p12 were the most relevant based on the number of interactions between proteins and the level of expression (Figure 4b). The p14-cluster contains annexin A2 protein (ANXA2_P07356) associated with several cellular functions, such as fibrinolysis, cell motility (in epithelial cells), protein related to actin cytoskeleton and cell matrix interactions [43], interacting with two other proteins involved in the intracellular lipid transport (FABP1_P12710 and FABP5_Q05816). Interestingly, another relevant group of proteins was identified in the p12-cluster with a very strong protein–protein interaction between several fibrinogens (FGL1_Q71KU9, FIBB_Q8K0E8, FIBA_E9PV24 and FIBGG_Q8VCM7). Fibrinogens are involved in hepatocyte growth, and they mediate blood platelet spreading, interstitial collagen and fibrotic lesions [44]. Furthermore, additional proteins were found to interact with fibrinogens (Appendix A), such as serum amyloid p-component (SAMP_P12246), hemopexin (HEMO_Q91X72) or hydroxyacid oxidase 2 (HAOX2_Q9NYQ2). Consistently with string results, ANXA2 and fibrinogen complexes were also the most enriched cellular components identified by FunRich analysis (Figure 4a). Interestingly, when comparing the data obtained by DDA and SWATH analysis, we confirmed that several proteins identified by SWATH were also identified by DDA analysis, including several related to fibrinogen complex proteins (SAMP/*Apcs* and S10A9/*S100a9*; Appendix A).

### 3.4. Validation of SWATH–MS Analysis Unmasks the Fibrinogen Complex as a New Molecular Mechanism Related to PLD

Based on these data, we selected from the SWATH–MS analysis (Appendix A) the altered proteins related to the fibrinogen complex at p12 and p14 stages for in vivo validation with RT-qPCR, Western blotting and immunohistochemistry, in order to establish their confirmation as possible targets of disease. Eight out of 10 of these selected proteins were validated at mRNA level by RT-qPCR (Figure 5); one up-regulated at the 14-stage (ANXA2_P07356) and at p12-group (SAMP_P12246, FGL1_Q71KU9, ILK_O55222, S10A9_P31725, FIBB_Q8K0E8, HEMO_Q91X72, FIBA_E9PV24 and FIBG_Q8VCM7), and one down-regulated at the p12 group (HAOX2_Q9NYQ2). Interestingly, gene expression of the two proteins with more distance or fewer interactions to the fibrinogen group (ILK and S10A9) were the only ones that were not validated (Figure 5c).

Up-regulation of the fibrinogen complex (ANXA2, SAMP, FIBB, FIBA, FIBG and HAOX2 proteins) was further confirmed by *Western blotting* (WB) analysis (Figure 6). Overall, gene expression and WB were highly consistent with SWATH-MS proteomic data. Finally, we also performed immunohistochemistry validation of the fibrinogen complex in mutant and polycystic liver samples (Figure 7). Fibrinogen staining was strongly up-regulated in cyst-lining epithelial cells and bile duct dilatations (Figure 7a). These results unmask, for the first time, the fibrinogen complex as a possible pathway related to hepatic cystogenesis and as a possible future therapeutic target for PLD.

## 4. Discussion

Polycystic liver disease (PLD) is commonly an extrarenal manifestation associated with autosomal dominant polycystic kidney disease (ADPKD). PLD is characterized by progressive dilation of the bile ducts and progression of multiple cysts, occupying at least half the volume of the liver parenchyma. Several of the current therapeutic strategies are based on surgical and pharmacological procedures to improve the symptoms of the disease. For this reason, the treatment of PLD has been ineffective so far, and the only curative option is liver transplantation [45]. This is mainly because the key intrinsic molecular mechanism of cystogenesis remains unknown. The main approaches are focused on the cAMP signaling pathway and somatostatin analogues, and several new targets (for example, histone deacetylase 6, Cdc25A phosphatase, PPAR-γ and matrix metalloproteases) have been evaluated in preclinical studies, but need to be tested clinically [8]. So, an emerging interest in understanding molecular pathways and developing new therapeutic strategies is increasing [2,8]. The purpose of present and novel SWATH-MS work was characterized by the proteomic changes in polycystic livers and provided new insights and therapeutic targets for PLD.

In previous studies, we have identified in an orthologous model of ADPKD (*Pkd1^cond/cond^;Tam-Cre*) a developmental window for kidney cystogenesis, suggesting that timing of secondary events may influence the severity of cystic kidney disease. Using the same strategy, we investigated whether this window of renal development corresponded in time with a similar window of hepatic development. Interestingly, we observed that cystic liver phenotype does not follow the same timing, suggesting that kidney and liver have possible different mechanisms undergoing cystogenesis. Further studies should be focused on identifying the common and different mechanisms of disease in both organs, and identifying the unknown liver developmental window for cystogenesis. For further analysis, we used a next generation quantitative proteomic approach SWATH-MS to perform differential proteome analysis of cystic liver disease in a mild and more severe PLD phenotype within the same kidney developmental window. Our data suggest that in advanced stages of disease progression, the number of pathways deregulated are increased, probably due to secondary effects of the disease. In the same way, these results reinforce the idea that cyst growth passes with two big stages: cyst initiation and cyst progression [46]. Interestingly, both proteome analyses from a mild (p14 group) and a severe (p12 group) cystic status identified protein complexes of fibrinolysis directly related to liver cystic disease. At the milder stages of the disease, we identified ANXA2 as one of the interesting proteins to validate as a potential therapeutic target for PLD. ANXA2 is a calcium-regulated membrane-binding protein produced by a wide range of cell types, including epithelial, dendritic, trophoblast, and tumor cells, as well as monocytes and macrophages. The main role is to maintain cell surface proteolytic activity, but also fulfills a range of intracellular functions, including exocytosis, endocytosis, membrane repair [47] and maintenance of adherent-like intercellular junctions [48]. Its numerous functions contribute to fibrinolysis, regulation of inflammation and immune system activation, and tissue injury and repair; as a result, ANXA2 dysfunction has been implicated in multiple human diseases, for example, extracellular matrix remodeling and hepatic fibrosis [49,50], which is a common feature in disease progression of PLD [8]. Interestingly, at the severe stage of liver disease, our proteome analysis discovered a number of different proteins related to the fibrinogen complex. During tissue and vascular injury, fibrinogen is converted enzymatically by thrombin to fibrin, and during fibrinolysis, fibrin is degraded by the main enzyme plasmin (the active form of plasminogen). Fibrinogens are associated with MAPK [51,52] and integrins [53], two pathways that have been shown to be effective as potential therapy for ADPKD in preclinical studies [54,55]. Even the animal model of one of the overexpressed fibrinogens (FIBA) has been shown to have cystic disease [56]. Moreover, the increased activity of fibrinogen-clusters in PLD offers a new perspective to treat the disease in comparison to the current experimental drugs [57]. Consequently, the decrease in this cluster could support a new mechanism of actions for drugs in which fibrinogen family proteins would be the therapeutic targets. A greater example would be the blockade of ERRγ (estrogen related receptor γ), an orphan nuclear receptor [58,59], which modulates fibrinogen levels in hypofibrinogenemia states caused by diet-induced obesity, diabetes mellitus type 2, liver injury and alcohol-induced oxidative stress [60,61]. Specifically, inverse agonists, such as GSK5182 (CAS: 877387-37-6) [60,61,62,63,64,65,66] and DN200434 [67,68], would offer such beneficial action through the decrease of fibrinogen levels.

In summary, we identified several novel targets and pathways involved in physiopathology of PLD by quantitative proteomic SWATH-MS technology. We emphasized the novel therapeutic opportunity presented by fibrinogens and fibrinolysis (and the other related identified proteins), mainly due the fact that all validations with significant differences carried out are related to the fibrinogen complex. Nevertheless, there is potential to confirm this in future preclinical studies as well as more functional studies to characterize the exact role of our target candidates in PLD. In conclusion, this work has created a new mechanism and opportunity for future research in PLD physiopathology, leading to possible new therapeutic approaches of the disease.

## 5. Summary

We reported a novel proteomic quantitative study based on SWATH-MS technology comparing proteomes of *Wild Type* and polycystic livers. Several novel pathways and targets were identified, expanding the knowledge about molecular mechanisms in the PLD field and highlighting the fibrinogen complex as a possible new therapeutic target.

## Figures and Tables

**Figure 1 biomedicines-10-00290-f001:**
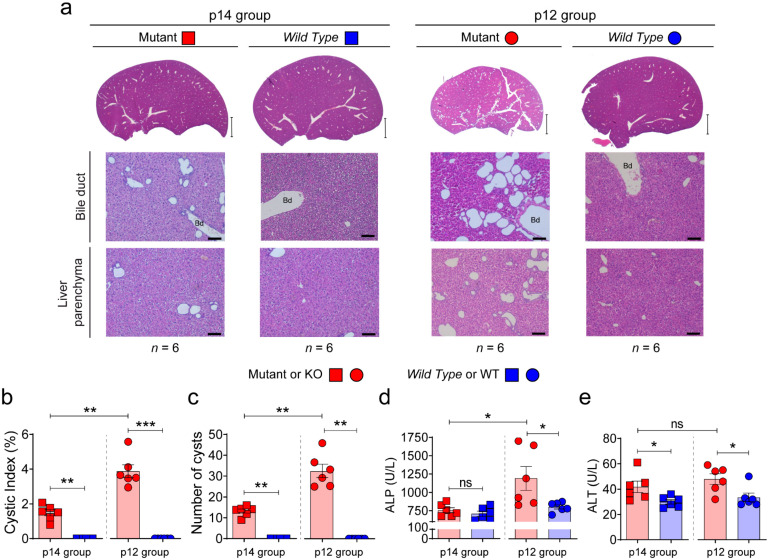
Characterization of the cystic liver phenotype at day 30 form p12 and p14 stages. *Pkd1* deletion at p12 carries a more severe cystogenesis. (**a**) Representative macroscopic and microscopic images of hematoxylin–eosin stained from livers of *Pkd1^cond/cond^;Tam-Cre* mouse *Wild Type* (WT, *Pkd1^cond/cond^;Tam-Cre^−^*) and Mutant (KO, *Pkd1^cond/cond^;Tam-Cre^+^*) with *Pkd1* gene deletion induced by tamoxifen at postnatal day 14 (p14 group) and 12 (p12 group). All mice were sacrificed at p30. Scale bar, 2 mm (upper panel) 100 µm (lower panel). *n* represents the number of samples per group, and Bd means bile duct. (**b**,**c**) Hepatic cystic index and number of cysts of the different phenotypes. (**d**,**e**) Blood serum ALP and ALT values. Bars represent means ± SEM in all cases. *p* < 0.05 by Student’s *t*-test (two-tail) was considered as a significant result. ns represents not significance. * *p* < 0.05, ** *p* < 0.01, *** *p* < 0.001.

**Figure 2 biomedicines-10-00290-f002:**
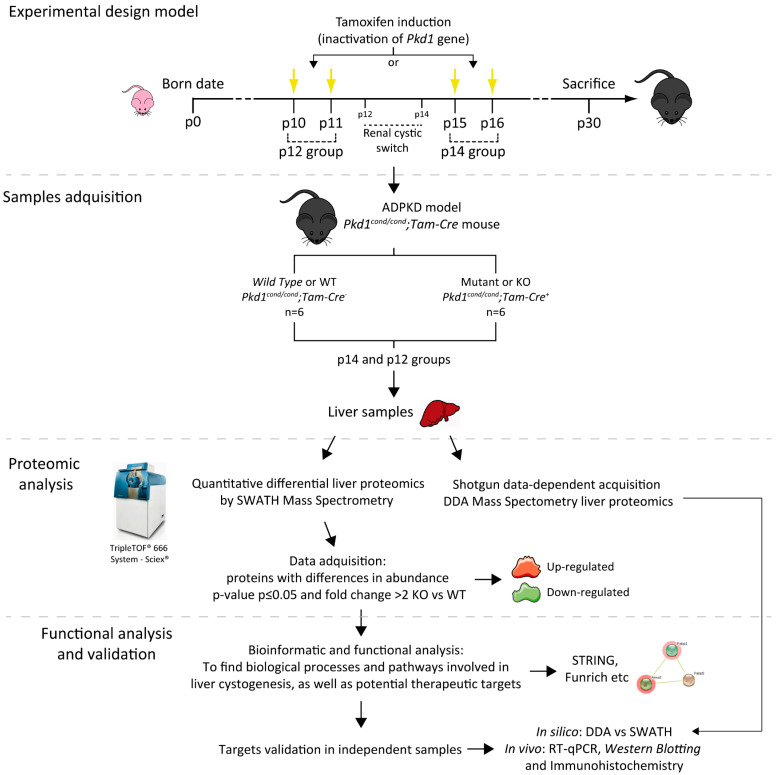
Illustrated workflow scheme. We used an ADPKD murine model (*Pkd1^cond/cond^;Tam-Cre* mice) in which the genetic inactivation of *Pkd1* was induced by tamoxifen administration at postnatal day 10 (p10) and p11 (rapid disease progression) or p15 and p16 (delayed disease progression) according to the developmental switch for renal cystogenesis, defining the two groups of the study named p12 and p14 respectively [19]. For both groups, we used an *n* equal or greater than six *Wild Type* (WT) and Mutant (KO) individuals in each condition, and all animals were sacrificed at p30. We studied the differential proteome of WT and KO livers in both severities of liver cystic disease (Figure 1), using both quantitative proteomic SWATH–MS and shotgun data-dependent acquisition DDA-MS analysis. Of the proteins with a significant change in abundance, we used bioinformatics tools to detect novel therapeutic targets and pathways involved in disease. Finally, we validated those targets using first in silico (DDA vs SWATH analysis) and finally in vivo strategies, as well as RT-qPCR, *Western blotting* and immunohistochemistry. *p* means postnatal day.

**Figure 3 biomedicines-10-00290-f003:**
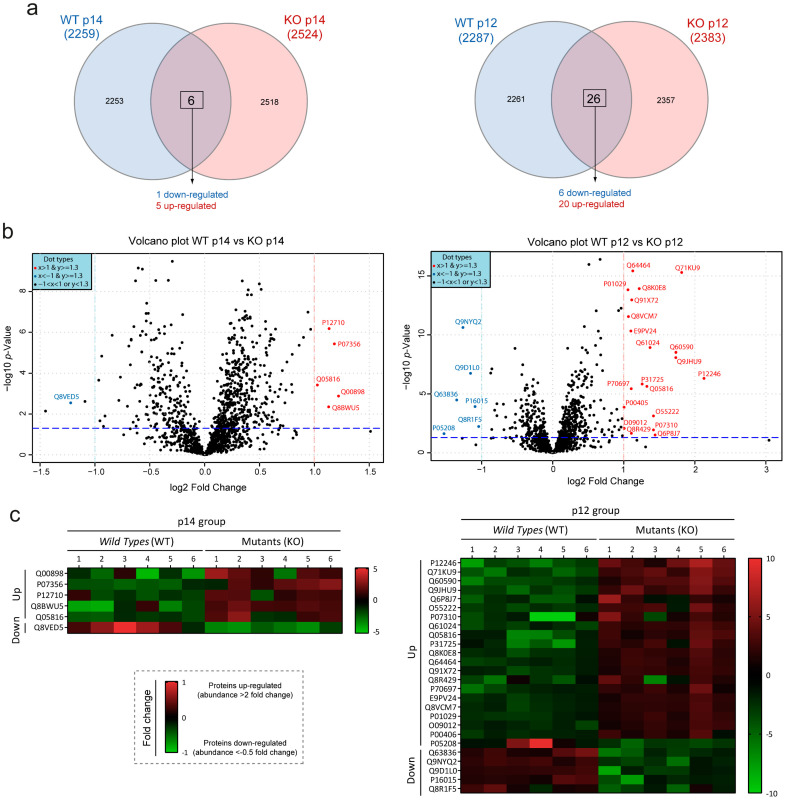
Graphic representation of proteins with altered abundance in hepatic cystogenesis. (**a**) Quantitative Venn diagrams showing the proteins with significant alterations (up- and down-regulated) of p14 (**left**) and p12 (**right**) groups. See Table 1 for the complete list of proteins with significant differences. (**b**) Volcano plots comparing *Wild Type* and Mutant samples in p14 (**left**) and p12 (**right**) groups, respectively. Volcano plots showing proteins with significant differences of protein abundance (*p*-value < 0.05 and fold change > 2) up-regulated (red dots) or down-regulated (blue dots) in SWATH analysis. X-axis shows the fold change of log(2) (the red and green lines show the 2-fold cut-off up- and down-regulated, respectively) and Y-axis shows −log(10)-*p*-value (blue lines shows the cut-off point of a *p*-value < 0.05). (**c**) The heat maps represent the significant differences in protein abundance between individuals according to the areas of the SWATH spectral library. Red represents the up-regulated and green the down-regulated proteins.

**Figure 4 biomedicines-10-00290-f004:**
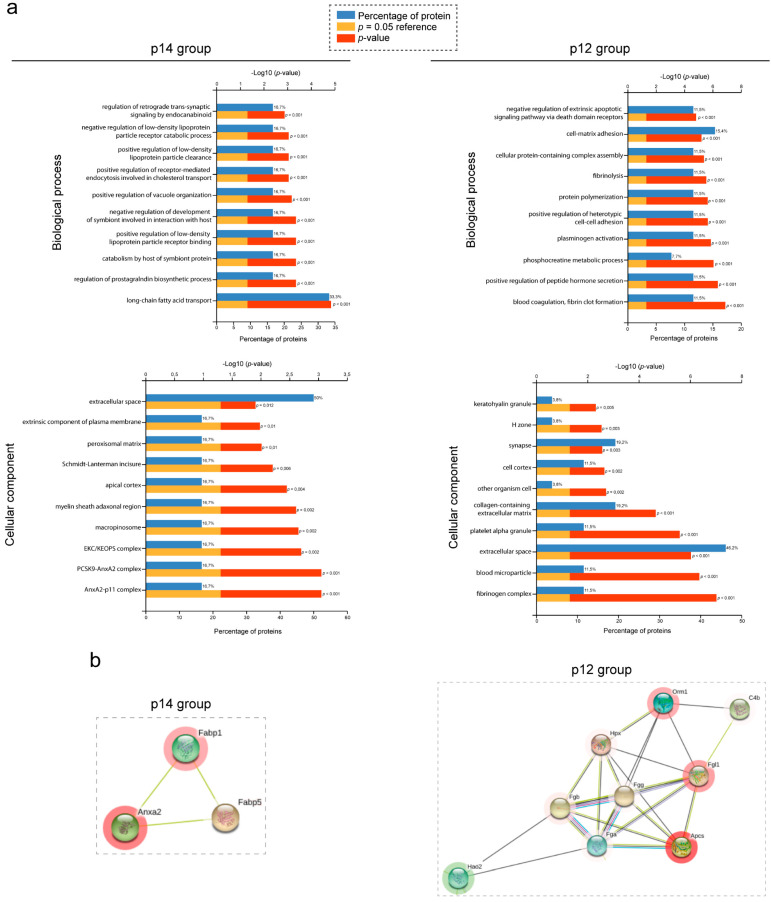
Functional analysis of PLD proteome. (**a**) FunRich functional enrichment analysis results. Proteins identified with proteomic quantified SWATH analysis in each group, p14 (left panel) and p12 (right panel), were submitted to biological process (upper panel) and cellular component (lower panel) in FunRich software. For all graphs, upper X-axis represents −log10 (*p*-value) and lower X-axis represents the percentage of proteins. The highest FDR and *p*-value top 10 process and components are shown, which were classified according to their *p*-value (down the lowest *p*-value). (**b**) Protein–protein interaction map of selected clusters according to string. Proteins with significant up- and down-regulated protein abundance were submitted to string analysis. Networks represent protein–protein analysis of selected and the most important clusters of p14 (**right**) and p12 (**left**) data, respectively. Proteins are represented as nodules, which are colored red or green imply up-regulate or down-regulate in protein abundance respectively.

**Figure 5 biomedicines-10-00290-f005:**
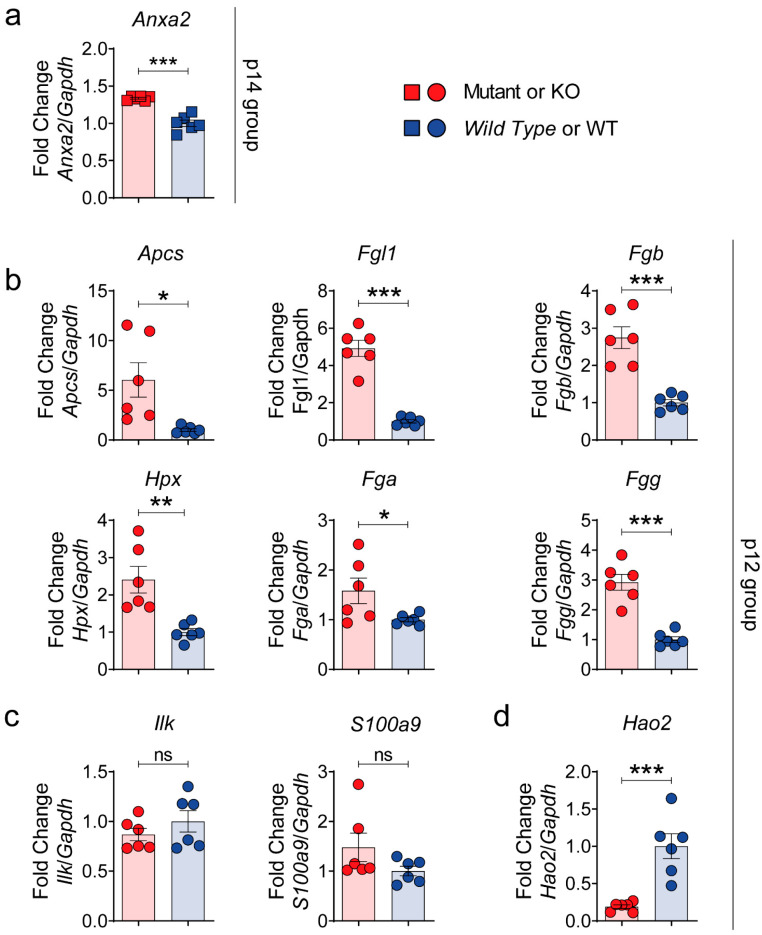
Validation of possible PLD targets by RT-qPCR. Gene expression of 8 to 10 selected targets correlated with SWATH-MS data. RT-qPCRs of up-regulated selected p14 target annexin A2 (*Anxa2*) (**a**); up-regulated p12 targets amyloid *p* component, serum (*Apcs*, SAMP gene), fibrinogen-like 1 (*Fgl1*), fibrinogen beta chain (*Fgb*), hemopexin (*Hpx*), fibrinogen alpha chain (*Fga*), fibrinogen gamma chain (*Fgg*) (**b**), integrin linked kinase (*Ilk*) and S100 calcium binding protein A9 (*S100a9*) (**c**); and down-regulated p12 target hydroxyacid oxidase 2 (*Hao2*) (**d**). *n* = 6 was used for each protein group. *Gapdh* was used as a housekeeping gene. Bars represent means ± SEM. Student’s *t*-test with two-tail was used and a value of *p* < 0.05 was considered significant. ns: not significant (*p* ≥ 0.05), * *p* < 0.05, ** *p* < 0.01, *** *p* < 0.001.

**Figure 6 biomedicines-10-00290-f006:**
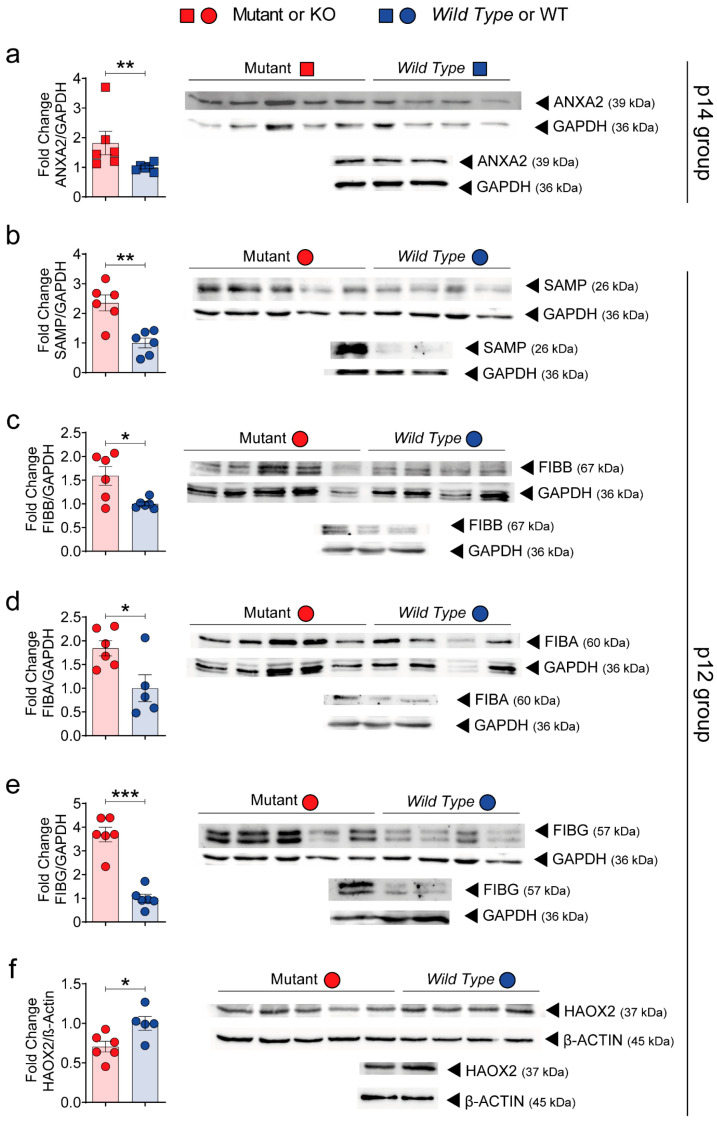
Validation of possible targets by *Western blotting*. The protein abundance of five possible targets of our quantitative proteomic SWATH-MS data were validated by *Western blotting* (WB) analysis. WB of up-regulated selected p14 target annexin A2 (ANXA2) (**a**); up-regulated p12 targets amyloid *p* component, serum (SAMP) (**b**), fibrinogen beta chain (FIBB) (**c**), fibrinogen alpha chain (FIBA) (**d**), fibrinogen gamma chain (FGG) (**e**); down-regulated p12 target hydroxyacid oxidase 2 (HAOX2) (**f**). *n* = 6 was used for each protein group. GAPDH or β-ACTIN were used as housekeeping genes. Bars represent means ± SEM. Student’s *t*-test with two-tail was used and a value of *p* < 0.05 was considered significant. * *p* < 0.05, ** *p* < 0.01, *** *p* < 0.001.

**Figure 7 biomedicines-10-00290-f007:**
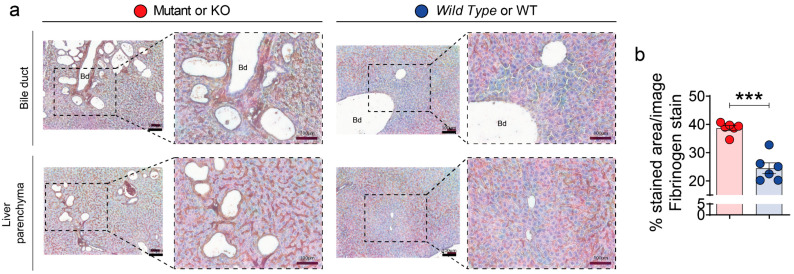
Fibrinogen complex is up-regulated in the cystic epithelia. (**a**,**b**) Representative images of fibrinogen immunohistochemistry in bile duct (upper panel) and liver parenchyma (lower panel) (**a**) and its quantification (**b**). *n* = 6 was used for each group. The immunohistochemistry analysis showed an up-regulation of fibrinogen through cystic epithelia. The samples corresponded to p12 group. Scale bars represents 100 µm. Bd means bile duct. Bars represent means ± SEM. Student’s *t*-test with two-tail was used and a value of *p* < 0.05 was considered significant. *** *p* < 0.001.

**Table 1 biomedicines-10-00290-t001:** List of proteins that presented an abundance with two-fold up-regulated (>2 fold-change) or down-regulated (<0.5 fold-change) and a significant adjusted *p*-value (according to parametric Student’s *t*-test) comparing *Wild Type* and Mutant polycystic livers. Table shows the list corresponding to the results with inactivation of the *Pkd1* gene at postnatal day 14 and 12. First, second and third columns represented the protein accession, protein code and gene name respectively. Fifth and sixth column represent the fold-change and *p*-value of quantitative SWATH analysis.

Protein Accession ^1^	Protein Code	Gene Name	Fold-Change WT to Mutant	*p*-Value (*p* ≤ 0.05 Student’s *t*-Test)
*Pkd1* deletion at postnatal day 14 (p14 group)				
Q00898	A1AT5	*Serpina1e*	2.33	1.32 × 10^−3^
P07356	ANXA2	*Anxa2*	2.27	3.72 × 10^−6^
P12710	FABPL	*Fabp1*	2.20	6.58 × 10^−7^
Q8BWU5	OSGEP	*Osgep*	2.19	4.45 × 10^−3^
Q05816	FABP5	*Fabp5*	2.04	3.76 × 10^−4^
Q8VED5	K2C79	*Krt79*	0.43	2.82 × 10^−3^
*Pkd1* deletion at postnatal day 12 (p12 group)				
P12246	SAMP	*Apcs*	4.37	4.79 × 10^−7^
Q71KU9	FGL1	*Fgl1*	3.52	5.02 × 10^−16^
Q60590	A1AG1	*Orm1*	3.33	2.99 × 10^−9^
Q9JHU9	INO1	*Isyna1*	3.32	8.19 × 10^−9^
Q6P8J7	KCRS	*Ckmt2*	2.71	3.01 × 10^−2^
O55222	ILK	*Ilk*	2.67	7.46 × 10^−4^
P07310	KCRM	*Ckm*	2.67	1.13 × 10^−2^
Q61024	ASNS	*Asns*	2.58	1.21 × 10^−9^
Q05816	FABP5	*Fabp5*	2.51	2.28 × 10^−6^
P31725	S10A9	*S100a9*	2.39	1.45 × 10^−6^
Q8K0E8	FIBB	*Fgb*	2.33	1.21 × 10^−14^
Q64464	CP3AD	*Cyp3a13*	2.18	3.77 × 10^−16^
Q91X72	HEMO	*Hpx*	2.16	1.08 × 10^−13^
Q8R429	AT2A1	*Atp2a1*	2.15	2.20 × 10^−2^
P70697	DCUP	*Urod*	2.15	3.57 × 10^−6^
E9PV24	FIBA	*Fga*	2.14	4.78 × 10^−11^
Q8VCM7	FIBG	*Fgg*	2.09	2.84 × 10^−12^
P01029	CO4B	*C4b*	2.08	1.51 × 10^−14^
O09012	PEX5	*Pex5*	2.01	8.43 × 10^−3^
P00405	COX2	*Mtco2*	2.01	1.35 × 10^−4^
P05208	CEL2A	*Cela2a*	0.35	2.37 × 10^−2^
Q63836	SBP2	*Selenbp2*	0.39	3.27 × 10^−5^
Q9NYQ2	HAOX2	*Hao2*	0.42	2.40 × 10^−11^
Q9D1L0	CHCH2	*Chchd2*	0.45	1.79 × 10^−7^
P16015	CAH3	*Ca3*	0.47	1.19 × 10^−4^
Q8R1F5	HYI	*Hyi*	0.49	5.79 × 10^−3^

^1^ Protein accession according to Uniprot [38].

## Data Availability

The data presented in this study are available in this article and this Appendix A. In addition, The mass spectrometry proteomics data have been deposited to the ProteomeXchange Consortium via the PRIDE [69] partner repository with the dataset identifier PXD031253.

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
