# Peer review of "Quantitative Proteomic Study Unmasks Fibrinogen Pathway in Polycystic Liver Disease"

_biomedicines, 2022, doi:10.3390/biomedicines10020290_

Round 1

Reviewer 1 Report

In this manuscript, Cordido and collaborators analyzed the liver of a mouse model of Polycystic Kidney Disease (PKD), used in this case a a model of Polycystic Liver Disease (PLD), with the aim of identifying novel pharmacological targets for the therapeutic management of PLD. They performed a quantitative proteomic analysis by the SWATH‐MS technology. 

Main issues

The authors should better justify the choice of the animal model, making clear whether Pkd1cond/cond;Tam-Cre-/+ mice have been used in other studies as models of PLD and adding references in the Introduction to confirm the translational level of their results. 

The mRNA and protein expression they evaluated after the proteomic analysis cannot be defined a real "target validation". To perform a real validation, they should treat animals with either agonists or antagonists of their targets and check their efficacy. Please discuss this point and rephrase. 

The paper requires a careful revision of the language, if possible by an English mother tongue. 

Minor issues

Please report additional information in the Table of the primer used for qPCR (RefSeq and product length)

Please check the resolution of the Figures, which is in some cases not sufficient. 

Please remove the legend for p value (** etc), that should be present only in the Figure legends. 

Author Response

Dear Dr. Terence Chen,

We were very pleased to see that the reviewers were enthusiastic about the developed study, although they had requested more information about some specific aspects. We agree that including this information would strengthen the manuscript and thus we have addressed and performed all the requested suggestions and corrections. We also wish to thank the two reviewers for their helpful comments and suggestions to improve our manuscript.

Enclosed please find our “point-by-point” responses to the specific questions raised by the reviewers. I have listed individually the comments/criticisms of each reviewer in black, followed by our response in blue.

Reviewer 1

-In this manuscript, Cordido and collaborators analyzed the liver of a mouse model of Polycystic Kidney Disease (PKD), used in this case a model of Polycystic Liver Disease (PLD), with the aim of identifying novel pharmacological targets for the therapeutic management of PLD. They performed a quantitative proteomic analysis by the SWATH‐MS technology.

Main issues

1) The authors should better justify the choice of the animal model, making clear whether Pkd1cond/cond;Tam-Cre-/+ mice have been used in other studies as models of PLD and adding references in the Introduction to confirm the translational level of their results.

We thank Reviewer#1 comment and we have followed their suggestion. Different animal models have been useful for the improvement in the knowledge of Polycystic Kidney and Liver Diseases (PKD and PLD, respectively). In fact, there is no specific, optimal and unique murine model for PKD diseases. This fact is highlighted by the huge amount of models in current research [1–8], among others. One of the most relevant differences among all the models is the possible timing and appearance of cysts in kidney and/or liver upon  inactivation of PKD genes. In consequence, the selection of each animal must be carefully assessed in order to achieve the desired goal.

The main aim of our research is to deepen the knowledge of hepatic manifestations regarding PKD diseases. Therefore, we need a rodent model whose hepatic phenotype must be constant and fully penetrant. Hence, the model chosen must manifest hepatic cysts and bile duct dilatation as well as clinical conditions similar to ADPKD patients. Pkd1cond/cond;Tam-Cre-/+ model have these requirements, because all of the mutant animals (Pkd1cond/cond;Tam-Cre+) have hepatic cysts and aberrant hepatic function, shown by biochemical studies; whereas Wild Type animals (Pkd1cond/cond;Tam-Cre+) are healthy. Additionally, the use of conditional rodent models over germinal models has another beneficial aspect. In this article we show for the first time that the clinical manifestations of the hepatic phenotype are strongly dependent on the time of inactivation of PKD genes. In consequence, an earlier inactivation shows a worse phenotype and progression of disease. However, if the inactivation takes place at a later point the progression and clinical signs of the diseases are milder. In order to look for pathways involved in the pathophysiology of the PLD as well as to seek pharmacological treatments, the studies of both severe and mild conditions are essential. For this reason, this model could be the basis for future studies in which the biological basis of PLD diseases could be dissected.

Taking into account the comments of the Reviewer#1, we have expanded the explanation of the use of our animal model in the introduction with the pertinent references (Lines 54-60, Page 2). In addition, we have expanded this justification raised by Reviewer#1 in the first part of the results section 3.1. (Lines 283-303, pages 6-7).

  1. Garcia-Gonzalez, M.A.; Menezes, L.F.; Piontek, K.B.; Kaimori, J.; Huso, D.L.; Watnick, T.; Onuchic, L.F.; Guay-Woodford, L.M.; Germino, G.G. Genetic interaction studies link autosomal dominant and recessive polycystic kidney disease in a common pathway. Hum. Mol. Genet.2007, 16, 1940–1950, doi:10.1093/hmg/ddm141.
  2. Lager, D.J.; Qian, Q.; Bengal, R.J.; Ishibashi, M.; Torres, V.E. The pck rat: A new model that resembles human autosomal dominant polycystic kidney and liver disease. Kidney Int.2001, 59, 126–136, doi:10.1046/J.1523-1755.2001.00473.X.
  3. J L Fry Jr, W E Koch, J C Jennette, E McFarland, F A Fried, J.M. A genetically determined murine model of infantile polycystic kidney disease. J. Urol.1985, 184, 828–33.
  4. Takahashi, H.; Calvet, J.P.; Dittemore-Hoover, D.; Yoshida, K.; Grantham, J.J.; Gattone, V.H. A hereditary model of slowly progressive polycystic kidney disease in the mouse. J. Am. Soc. Nephrol.1991, 1, 980–989, doi:10.1681/ASN.V17980.
  5. Outeda, P.; Menezes, L.; Hartung, E.A.; Bridges, S.; Zhou, F.; Zhu, X.; Xu, H.; Huang, Q.; Yao, Q.; Qian, F.; et al. A novel model of autosomal recessive polycystic kidney questions the role of the fibrocystin C-terminus in disease mechanism. Kidney Int.2017, 92, 1130–1144, doi:10.1016/J.KINT.2017.04.027.
  6. Gallagher, A.R.; Esquivel, E.L.; Briere, T.S.; Tian, X.; Mitobe, M.; Menezes, L.F.; Markowitz, G.S.; Jain, D.; Onuchic, L.F.; Somlo, S. Biliary and pancreatic dysgenesis in mice harboring a mutation in Pkhd1. Am. J. Pathol.2008, 172, 417–429, doi:10.2353/AJPATH.2008.070381.
  7. Wodarczyk, C.; Rowe, I.; Chiaravalli, M.; Pema, M.; Qian, F.; Boletta, A. A novel mouse model reveals that polycystin-1 deficiency in ependyma and choroid plexus results in dysfunctional cilia and hydrocephalus. PLoS One2009, 4, doi:10.1371/JOURNAL.PONE.0007137.
  8. Baltimore PKD Center | Mouse Models / Biobank Core Available online: http://www.baltimorepkdcenter.org/mouse/index.shtml (accessed on Jun 21, 2019).

2) The mRNA and protein expression they evaluated after the proteomic analysis cannot be defined a real "target validation". To perform a real validation, they should treat animals with either agonists or antagonists of their targets and check their efficacy. Please discuss this point and rephrase.

We completely agree with Reviewer#1 that this can lead to misinterpretation, and we have changed “target validation” for “SWATH data validation of possible targets”.  We also agree with Reviewers statement of using agonist and antagonist would be the final prove of possible diseases targets, as we mentioned in the last paragraphs of the discussion (lines 555-564, page 17).  Indeed, our future efforts are focused on this validation, and we strongly believe that we are going to be pleased to share our latest discoveries in this topic soon.

The paper requires a careful revision of the language, if possible by an English mother tongue.

We have followed the feedback from both reviewers on this issue and have submitted the article for correction by a native English speaker.

Minor issues

Please report additional information in the Table of the primer used for qPCR (RefSeq and product length)

We thank Reviewer#1 for this comment. We have completed Table S2 with the requested information: RefSeq of transcript sequencings used according to Nucleotide ID and the amplicon length.

Please check the resolution of the Figures, which is in some cases not sufficient.

We thank Reviewer#1 for this comment which has also been addressed by Reviewer#2. We agree, the resolution of the images is not adequate and we have improved it.

Please remove the legend for p value (** etc), that should be present only in the Figure legends.

Thanks again for the comment. We have removed the information requested and we have corrected all these kinds of errors on the manuscript.

Reviewer 2 Report

-In this manuscript, the authors researched on “Quantitative proteomic study unmasks fibrinogen pathway in Polycystic Liver Disease. The authors reported that molecular pathway in cystic liver disease highlighting the fibrinogen complex. This paper can be interesting in liver disease domain. I affirm its acceptance for publication. However, there are some concerns before it is acceptable.

Comments:

Q1: Author could revise the keywords section. It is not uniform and add more keywords.

Q2: In line 101, -80 ºC, it is not properly marked. Author could revise it in full files.

Q3: In line 104-105, % and space issues are there, it needs to be revise.  It is not impressive and revise whole manuscript file.

Q4, In line 362, Author should explain how PCA data can analyzed for KO p12 and WT p12? In figure S6, what type of software used? PCA abbreviation is not unform.

Q5: Significance of various liver diseases associated genes, proteins, and metabolites have been recently documented. I encourage below research articles to cite on this manuscript. They very recently discussed various part of liver disease.   

https://doi.org/10.3390/ijms22031160; https://doi.org/10.3390/ijms22158309; https://doi.org/10.3390/ijms22126326; https://doi.org/10.3390/cells10102634; https://doi.org/10.3390/microorganisms9020296   

Q6:  Line no 150 (page 4), 100TOF MS/MS- the sentence space issues found in line no 150.  Author should revise full file

Q7:  In summary, author should revise with list of significant proteins and genes expressions.  

Q8: in figure S5, Cluster analysis of proteins lines (left side) are not good. Author should improve the figure resolution.

Q9, Language, sentence space, technical errors correction must be focus on this manuscript draft.

Author Response

Dear Dr. Terence Chen,

We were very pleased to see that the reviewers were enthusiastic about the developed study, although they had requested more information about some specific aspects. We agree that including this information would strengthen the manuscript and thus we have addressed and performed all the requested suggestions and corrections. We also wish to thank the two reviewers for their helpful comments and suggestions to improve our manuscript.

Enclosed please find our “point-by-point” responses to the specific questions raised by the reviewers. I have listed individually the comments/criticisms of each reviewer in black, followed by our response in blue.

Reviewer#2

-In this manuscript, the authors researched on “Quantitative proteomic study unmasks fibrinogen pathway in Polycystic Liver Disease. The authors reported that molecular pathway in cystic liver disease highlighting the fibrinogen complex. This paper can be interesting in liver disease domain. I affirm its acceptance for publication. However, there are some concerns before it is acceptable.

Comments:

Q1: Author could revise the keywords section. It is not uniform and add more keywords.

We thank Reviewer#2 for this comment. We have revised the keywords section to be consistent with the abstract.

Q2: In line 101, -80 ºC, it is not properly marked. Author could revise it in full files.

We agree with Reviewer#2. We have removed the information requested and we have corrected all these typo errors on the manuscript.

Q3: In line 104-105, % and space issues are there, it needs to be revise.  It is not impressive and revise whole manuscript file.

We thank Reviewer#2 for this comment. We have revised the information requested and we have corrected all these kinds of typos on the manuscript.

Q4, In line 362, Author should explain how PCA data can analyzed for KO p12 and WT p12? In figure S6, what type of software used? PCA abbreviation is not unform.

We have followed Reviewer#2 suggestions and we included a brief explanation of KOp12 vs WTp12 PCA analysis (Please see Page 9, lines 366-370), and additionally to the original reference of the type of software mentioned on section Material and Methods (please see, Page 4, line 173-188), we also included the software used on Figure Legend S6 for easy and comprehensive reading. PCA abbreviation have been uniformed across the article.

Q5: Significance of various liver diseases associated genes, proteins, and metabolites have been recently documented. I encourage below research articles to cite on this manuscript. They very recently discussed various part of liver disease.  

We thank Reviewer#2 for this comment. We have added some of the proposed references in the Discussion section (Line 545, Page 16).

Q6:  Line no 150 (page 4), 100TOF MS/MS- the sentence space issues found in line no 150.  Author should revise full file

We thank Reviewer#2 for this comment. We have revised the information requested and we have corrected this typo on the manuscript.

Q7:  In summary, author should revise with list of significant proteins and genes expressions. 

In order to address Reviewer#2 comment, we have completed the supplementary material with a table where we summarized all data from SWATH, RT-qPCR and Western Blotting (please see Table S4, page 37). In this table it is possible to check all Fold Changes and p-values of all possible targets from the three data.

Q8: in figure S5, Cluster analysis of proteins lines (left side) are not good. Author should improve the figure resolution.

This is a formatting error, but we agree with Reviewer#2 and #1, and we have improved image resolution for all figures.

Q9, Language, sentence space, technical errors correction must be focus on this manuscript draft.

We have followed Reviewer# 1 and #2 suggestion and submitted the article for correction by a native English speaker. In addition to the typo identified by the reviewers, an exhaustive revision was done.

Round 2

Reviewer 1 Report

The authors have addressed the issues raised by the Reviewers. The manuscript is now suitable for publication. 

Author Response

Dear Reviewe#1. Thank you very much for helping us to improve our article. 

Best regards,

Miguel A García-Gonzalez